# Do non-pharmaceutical policies in response to COVID-19 affect stock performance? Evidence from Malaysia stock market return and volatility

**Racquel Rowland[1], Ricky Chee Jiun Chia[2]\*, Venus Khim-Sen Liew[1]**

**1** Universiti Malaysia Sarawak, Kota Samarahan, Malaysia, **2** Universiti Malaysia Sabah, Kota Kinabalu, Malaysia

\* ricky_82@ums.edu.my

**Data Availability Statement:** All the datasets can be found in CEIC website. See: https://www.ceicdata.com/en.

## Abstract

This paper examines the impact of non-pharmaceutical intervention by government on stock market return as well as volatility. Using daily Malaysian equity data from January 28, 2020 to May 31, 2022, the regression analysis with bootstrapping technique reveals that the government's response in combating the deadly virus through Stringency index has shown a positive direct effect on both stock market returns and volatility, and indirect negative effect on stock market returns. The study revealed that international travel restriction and cancelling public events are the major contributors to the growth of volatility when estimated for Malaysia stock market index. On the one hand, heterogenous impact is expected from the perspective of different sectors when the individual social distancing measures were taken into account in determining stock return and volatility. Apart from that, the robustness check for the main findings remains intact in majority of the regression models after incorporating daily COVID-19 death rate, log (daily vaccination) and day-of-the-week effect as additional control variable in alternative.

## Introduction

The Coronavirus disease (COVID 19) is a newly discovered pneumonia-causing virus which originated from Wuhan, China has caused serious damage to the economy and in fact, it is the largest economic shock the world has witnessed in decades. This black swan event has put the global economy on halt due to its contagious virus and its ability to spread wide like a sporadic effect that leads to millions of deaths worldwide. Given the severity of transmission, World Health Organization (WHO) announced the outbreak of virus as a global pandemic on 11 March, 2020. It can be seen that the event threatened not only to human well-being but also the country economy. With the surge in the total confirmed cases worldwide, governments across countries enforced several measurements to combat the deadly disease from spreading with the hope that the infection and fatality rates would reduce. Strict policies such as social distancing measures, public awareness program, mass testing and quarantine policies as well

**Funding:** The authors would like to acknowledge the financial support from Universiti Malaysia Sabah, Project Code: SDK0157-2020, Dr. Ricky Chee Jiun Chia.

**Competing interests:** The authors have declared that no competing interests exist.

as travel restriction were put in place to minimize the infection risk. There are some countries that went into a lockdown mode to curb the deadly transmission as their healthcare supporting system was at the verge of collapsing. This is especially true for the developed countries. The downside of having total lockdown is that it could cause massive disruption in global business and economic activities, thus this uncertainty would eventually lead to stock market crash.

A week after the WHO's announcement regarding the pandemic, most of Asia's main markets remained in the red. For instances, Indonesia's Jakarta Composite Index fell 4.99%, while South Korea's Kospi was down 2.47%, and Singapore's Straits Times lost 1.65%. Malaysia was no excepted. FBMKLCI, which represents Malaysia's main market index witnessed a steep decline in March 2020 and dropped below 1,500-point territory. The index reached its trough on March 19, 2020 with 1,219.72 points and followed by a rebound after the government announced social distancing measures to contain the virus during the early stage of coronavirus outbreak. This can be evident from a substantial growth in the government stringency policy index in Fig 1. It is also acknowledged that the market index slowly picked up the pace and grew steadily above 1500-point territory due to the investors' expectation of economic resumption as the government has started relaxing the rules and allowed more businesses to resume operation after the daily diagnosis of Covid-19 was on a downward trend as depicted in Fig 2. Malaysia was one of the Asian countries with stricter government responses in dealing with coronavirus. The country experienced higher compliance with "social distancing" advice since the outbreak on January 24, 2020. The first Covid-19 confirmed case was originated from the three travellers from China who entered Malaysia through Singapore and since then the daily cases rose to three digits as it hit the second wave. Fortunate enough, Malaysia managed to slow down the rate to zero local transmission on July 1, 2020 because of strict government policy. Figs 1 and 2 demonstrate that as the stringency index rose above 70 points, the government managed to flatten the curve on the second wave. Another downtrend pattern of Covid-19 from January 2021 onwards can be seen as soon as the government implemented another partial lockdown. It was reported that the daily cases dropped from 6,000 to 1,000 cases approximately during the third wave. However, there is a noticeable remark from Fig 1 that shows although the Stringency index remained above 70 points from May 2020 onwards, the number of daily Covid-19 did not reduce, but in fact it had gotten worst. The highest Covid-19 case reported in July 2021 crossed 17,000-mark with cumulative death of 5,000 approximately as

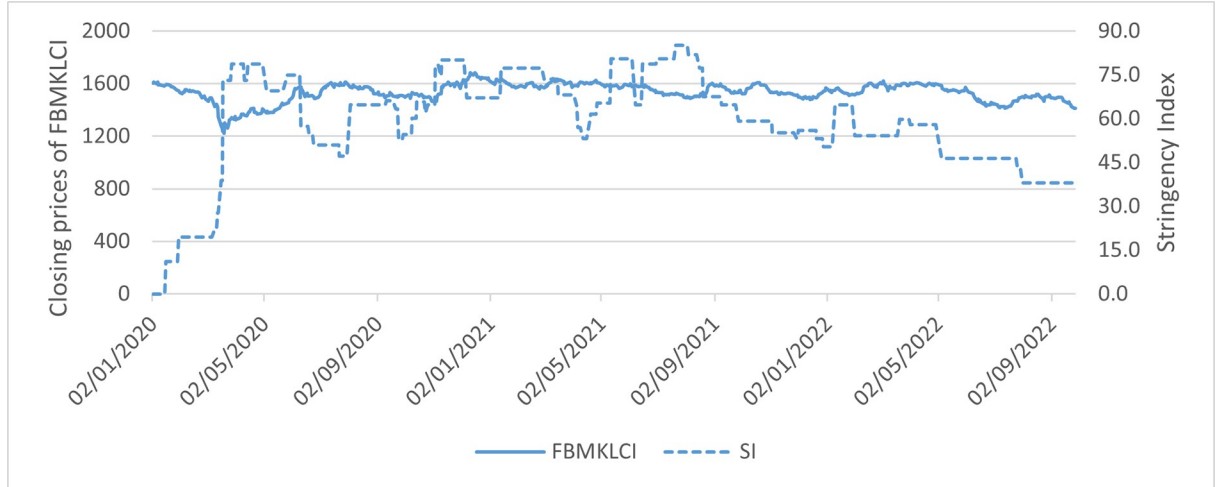

**Fig 1. The time series plot of FBMKLCI closing prices and Stringency index from January 2, 2020 to September 26, 2022.** Source: CEIC.

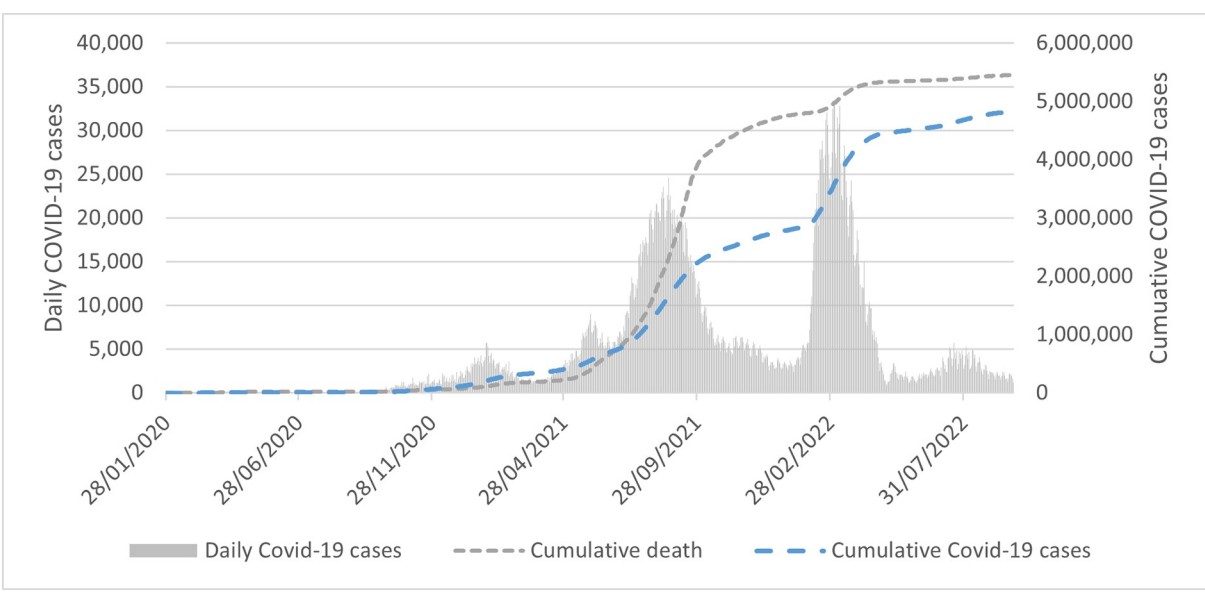

**Fig 2. The cumulative COVID-19 cases, cumulative death and daily COVID-19 cases in Malaysia from January 28, 2020 to September 26, 2022.** Source: CEIC.

portrayed in Fig 2. The second peak of daily confirmed COVID-19 cases can be spotted in the early year of 2022 with more than 30,000 cases are shown in Fig 2 despite the local market index maintaining its performance at 1400 to 1500-point territory. In the meantime, Fig 1 shows that the stringency index was below 70 points and continued to decline as the country started to loosen its social distancing measures domestically as well as internationally due to the fact that 77% of its population has been fully inoculated.

A significant amount of COVID-19 related study in financial market has been established since the outbreak of the disease in December 2019. Early studies such as Al-Awadhi et al. [1], Asraf [2] uncover that the focus on the pandemic-induced financial markets like daily growth rate and fatality rate of COVID-19 has significant effect on stock market return. Following the significant global impact, previous studies have documented a heterogenous effect across industries corresponding to the pandemic. For instance, Mazur, Dang and Vega [3] uncover that natural gas, food, healthcare and software stocks were the outperforming stocks whereas petroleum, real estate, entertainment and hospitality sectors were the underperforming stocks in US stock markets during the pandemic. Using event study approach, tourism (Liew [4]), transportation, mining, electricity & heating, and environment industries were adversely impacted by the pandemic whereas manufacturing, information technology, education and health care industries had been resilient based on the Chinese stock market (He et al. [5]). Other studies also investigated other class of asset as a safe haven investment in addition to the equities and this includes cryptocurrencies (Conlon, Corbet & McGee [6]; Demir et al. [7]), gold and oil prices (Salisu, Vo & Lawal [8]; Shaikh [9]).

Meanwhile, the association between non-pharmaceutical interventions and its impact on two financial market features (return and volatility) was rather limited and primarily focused on global perspective. In a recent study by Asraf [2], the empirical findings suggest that the announcement of social distancing by the government had a direct negative effect on the stock index of 77 selected countries, and an indirect positive effect on stock market returns as the investors were taking the reduction in Covid-19 cases as good news. Using stock return volatility of 67 countries, Zaremba et al. [10] further investigated the different types of non-

pharmaceutical policies and suggested that public information campaigns and cancellation of public events are the major contributors to the growth of volatility. In another study by Zaremba et al. [11], the role of non-pharmaceutical interventions in equity market liquidity was testified over 49 countries from both developed and emerging market which spanned from January to April 2020 using panel regression analysis. The study employed seven different policy responses namely school closures, workplace closures, cancelling public events, closing of public transportations, public information campaigns, restrictions on internal movement, and international travel control as explanatory variables, suggesting that public information campaign had positive and significant effect on liquidity. This can be explained that spreading information about the COVID-19 development may facilitate the pricing of a negative news about future states of economy in the stock market and thus, induce portfolio repositioning. Similarly, Chang, Feng and Zheng [12] employed panel data of 20 countries which cover the period of January 2 to July 21, 2020 for the dynamic panel model, suggesting that the overall government response, containment and health, and stringency indices have significantly positive effect on stock market returns. Specifically, government policy responses of shutting down workplaces, cancelling public events and restricting public gatherings and international travel, providing income support, and implementing fiscal measures could increase stock market returns.

Meanwhile, from the perspective of stock market volatility, Hunjra et al. [13] adopted Monte Carlo Simulation and found out that the public health measures and virus protection policies implemented in East Asian countries (China, Singapore, Thailand, and Japan) affect the capital market differently. It was reported that the volatility in Shanghai composite and FTSE Straits Time were heavily affected by the regulation concerning flight restriction. The highest volatility in Nikkei and SET was caused by the night curfew and social distancing policy, respectively. Baig et al. [14] employed different dimension of pandemic related data and suggested that Stringency index which proxied for social distancing measures by government elevated the US market volatility significantly. Bickley et al. [15] employed Hurst exponent method to measure the volatility persistence by comparing the pre and post policy exponents in conjunction with COVID-19 impact and found that stay-at-home policies elicited a stabilising response in market volatility across 28 countries in 6 continents.

Despite that, the impact of non-pharmaceutical intervention on different industries has been studied by several papers. For instance, Wang et al. [16] used three COVID-19 related government interventions namely stringency index, containment and health index and economic support index to analyze its impact on travel and leisure-related stock for nine major tourism destinations (United States, United Kingdom, France, Italy, Turkey, Denmark, Spain, Greece, Sweden) from January 2, 2020 to November 5, 2020. Based on quantile regression analysis, the study demonstrated that Stringency index which proxied for social distancing measure had positive(negative) and significant effect on travel and leisure-related stock return (volatility) and verified to be the most effective measures at mitigating the spread of COVID-19. A study conducted by Aldhamari et al. [17] illustrated that the abnormal returns of Malaysia stock market reacted negatively to the announcement of movement control order (MCO) over the 60 days of event windows using event study analysis. Further investigation shows that firms in the healthcare sector had significant positive cumulative average abnormal returns, with stock returns of the utilities and telecommunication firms showing no changes, while the remaining sectors fell remarkably. Using abnormal returns generated from event study as dependent variable, regression analysis revealed that the number of COVID-19 confirmed cases adversely affected firms' abnormal returns.

Hence, this study contributes to the existing literature in three aspects. First, the paper examines the effect of non-pharmaceutical policies using Stringency index on stock market

return and volatility in the context of Malaysia. The vaccination program was not fully ready until 24[th] February, 2021 and non-pharmaceutical strategy was the only way to curb the spread of deadly virus during that time. Thus, the policy implication from social distancing measures was examined in this study. Secondly, this study focuses on Malaysia stock market because Malaysia was among the first country to implement travel bans and lockdowns to combat the disease. With such government containment policies, it could potentially worsen Malaysia's economy. Although Malaysia's COVID-19 confirmed cases was relatively smaller during the first wave of pandemic as compared to other countries, the accumulated number of COVID-19 confirmed cases was accelerated at an exponential rate particularly during the third wave as the government started to ease the social distancing measure. The sudden spike in cases was due to weak compliance with COVID-19 SOPs which originated from two big contributors, the Benteng Lahad Datu cluster, in Sabah state and Kedah's Tembok cluster (Lai [18]). Thirdly, limited sample size of government response policies from previous studies has allowed the paper to estimate in detail the effect of individual sub-indices under Stringency index on the returns of stock market index as well as volatility for a longer timeframe. This is to verify whether such a policy would still be reliable for the market participants to restructure their portfolio in an unprecedented event. The remaining of the paper proceeds as follows: Section 2 briefly introduces data collection and methodology procedure. Section 3 discusses findings and finally, Section 4 concludes the study.

## Data and methodology

To investigate the implementation of non-pharmaceutical policy in response to the Covid-19 pandemic, this study employs stock market and pandemic-related data to demonstrate its impact on stock market return and volatility in the context of Malaysia. The study period starts on the trading day following Malaysia reported its first Covid-19 confirmed case on 25[th] January, 2020. The arrival of the news fell on Chinese New Year and stock market was closed on a public holiday. In consequence, the effect was only felt the next trading day which allows the sample to run from 28[th] January, 2020 to 31[st] May, 2022. For the most recent data collection, daily closing price of FBMKLCI is collected from CEIC website (https://www.ceicdata.com/en). Using similar source, the study collects pandemic-related data such as daily COVID-19 case and Stringency Index by searching the category under "Disease Outbreak" and "COVID-19 Economic Impact Indicator" respectively. The aforementioned index conveys different types of non-pharmaceutical interventions to curb the outbreak of pandemic and it is coded based on these 8 indicators which include school closing, workplace closing, cancel public events, restrictions gathering size, close public transport, stay at home requirements, restrictions internal movement and restriction on international. Each individual measure is aggregated and rescaled to obtain values from 0 to 100, where 0 for being the least stringent and 100 for being the most stringent policy responses. Macroeconomic variables such as inflation rate, interest rate, exchange rate, unemployment rate, and industrial production index are mainly collected from Department of Statistics Malaysia and Bank Negara Malaysia website (https://www.bnm.gov.my/) whereas the data of crude oil is downloaded from investing.com. Before proceeding with regression, the computation of FBMKLCI's return is illustrated in Eq (1):

$$RET_t = \left( \frac{P_t - P_{t-1}}{P_{t-1}} \right) \times 100 \tag{1}$$

where $RET_t$ is the stock return of FBMKLCI at time $t$, $P_t$ is the closing stock price at time $t$ and $P_{t-1}$ is the closing price at time $t-1$. Following the prior studies of Zaremba et al. [10], Asraf [2] and Baig et al. [14], this study has employed some variables from previous studies and

estimated the model specification using Ordinary Least Square (OLS) regression with boot-strapping technique. Hence, a new model specification is demonstrated in Eq (2).

$$Y_t = \alpha + \beta_1 CGRATE_t + \beta_2 SI_t + \sum_{c=1}^{C} \beta_c K_{c,t} + \varepsilon_t,$$ (2)

where $Y_t$ is the dependent variable for one of two variables, $RET_t$ and $VOL_t$ which measures in $|\log|RET|_t|$. $CGRATE_t$ refers to daily growth rate of COVID-19 confirmed cases which calculated as $(Cases_t - Cases_{t-1})/Cases_{t-1}$. The inclusion of daily growth rate of COVID-19 instead of the number of fatalities as the increasing number of confirmed cases due to coronavirus is associated with significant increase in both market returns and volatility. $SI_t$ denotes Stringency Index on day $t$ which proxied for non-pharmaceutical strategy and $K_t$ is a set of macro-economic variables which acts as control variables in the model. It is said that macroeconomic conditions have influenced on stock returns and volatility (See Bulmash and Trivoli [19]; Fama and French [20]; Hondroyiannis and Papapetrou [21]; Rapach, Wohor and Rangvid [22]; Corradi, Distaso, and Mele [23]; Salisu and Vo [24]; Lee, Lee and Wu [25]; and Uddin, Chowdhury, Anderson and Chaudhuri [26]). These macroeconomic variables include monthly data of Consumer Price Index which proxied for inflation rate (*INF*), rate return of Industrial Production Index (*RIPI*), overnight interbank policy which proxied for interest rate (*INT*) as well as unemployment rate (*UNEMPLOY*) and finally, daily rate return of USD/MYR exchange rate (*REXRATE*) and crude oil (*RCRUDE_OIL*). Since the highest frequency of data available for inflation rate, interest rate, unemployment rate and industrial production index is on a monthly basis, the study will allocate the data to each day in the particular month to which it relates.

Furthermore, to examine the indirect impact of non-pharmaceutical policy on stock market returns as well as stock volatility, the study extends Eq (2) by introducing interaction variable to the model specification as illustrated in Eq (3):

$$Y_t = \alpha + \beta_1 CGRATE_t + \beta_2 SI_t + \beta_3 (CGRATE_t \times SI_t) + \sum_{c=1}^{C} \beta_c K_{c,t} + \varepsilon_t,$$ (3)

where $CGRATE_t \times SI_t$ denotes the interaction term that used to estimate the value of coefficient $\beta_3$ in order to determine the stock market reaction to the growth of Covid-19 case through the government actions. The moderating effect is estimated from the variable explains that a restrictive policies can negatively influence investors' sentiment and increase market uncertainty but at the same time, the investors tend to appreciate the proactive measure taken by government in combating the virus or reducing the fatality rate and consequently, adjust their investment decision which leads to a positive market reaction.

Besides implementing the role of Stringency index which proxied for non-pharmaceutical policy as a whole, the study also determines to identify how individual government policy responses contribute to the stock return and volatility. Therefore, a new model specification is developed in Eq (4).

$$Y_t = \alpha + \beta_1 CGRATE_t + \sum_{j=1}^{J} \beta_j C_{j,t} + \sum_{c=1}^{C} \beta_c K_{c,t} + \varepsilon_t,$$ (4)

where $C_t$ represent eight sub-indices of containment and closure policies at time $t$ under the indicator of Stringency index. These policies include school closing (C1), workplace closing (C2), cancel public events (C3), restrictions on gatherings (C4), close public transport (C5), stay-at-home requirement (C6), restrictions on internal movement (C7) and international travel control (C8). All these variables are measured in ordinal scale and can be retrieved from CEIC database under the category of Oxford COVID-19 Government Response Tracker along with a detailed description.

**Table 1. Descriptive statistics.**

| Variables | $RET_{FBMKLCI}$ | $VOL_{FBMKLCI}$ | CGRATE | SI | C1 | C2 | C3 | C4 | C5 | C6 | C8 | INT | INF | RIPI | EX | UNEMPLOY | ΔCRUDE OIL |
|---|---|---|---|---|---|---|---|---|---|---|---|---|---|---|---|---|---|
| Mean | 0.00 | 0.68 | 2.71 | 62.20 | 2.14 | 1.99 | 1.79 | 3.04 | 0.47 | 1.15 | 2.72 | 1.88 | 122.29 | 0.03 | 0.01 | 4.47 | 0.11 |
| Std. Dev. | 0.96 | 0.67 | 9.45 | 15.48 | 0.98 | 0.89 | 0.53 | 1.37 | 0.59 | 0.93 | 0.64 | 0.28 | 2.28 | 9.44 | 0.31 | 0.44 | 3.53 |
| Minimum | -5.40 | 0.00 | 0.00 | 11.11 | 0.00 | 0.00 | 0.00 | 0.00 | 0.00 | 0.00 | 0.00 | 1.75 | 117.60 | -35.67 | -1.90 | 3.20 | -27.98 |
| Maximum | 6.63 | 6.63 | 180.71 | 85.19 | 3.00 | 3.00 | 2.00 | 4.00 | 2.00 | 3.00 | 4.00 | 2.75 | 126.60 | 23.53 | 2.19 | 5.30 | 19.08 |

This table presents descriptive statistics of different variables used in the empirical analysis of the study. $VOL_{FBMKLCI}$ represents volatility which is the logarithm of absolute daily returns of FBMKLCI ($|Log|RET|_t|$). RET is daily market return of FBMKLCI. The daily growth rate of Covid-19 is expressed as CGRATE. SI is the Stringency Index which measured on a scale of 0 to 100 that represents the daily government response to COVID-19 by using non-pharmaceutical policies and eight sub-indices under this indicator are school closing (C1), workplace closing (C2), cancel public events (C3), restrictions on gathering (C4), close public transport (C5), stay-at-home requirement (C6), restrictions of internal movement (C7) and international travel controls (C8). CRUDE OIL is the daily price of brent crude oil, EXRATE is the daily exchange rate of USD/MYR, UNEMPLOY is the monthly rate of unemployment. IPI is the monthly index of industrial production. INT is the monthly overnight policy rate.

## Results

Table 1 presents the descriptive statistic of the variables used in the study. It can be observed that stock return volatility was peaked at 0.84 during the pandemic. On the other hand, the maximum daily FBMKLCI's return is 6.63% and the minimum is -5.26%. It is also noted that the daily growth rate of Covid-19 has increased by 2.71%-5.40% on average. Meanwhile, the minimum (maximum) value of Stringency index is 11.11 (85.19) point. A low (high) level of stringency index is an indication of less (more) stringent measure being imposed to combat the deadly virus. Similarly, the minimum and maximum values of individual government measures show changes in policies in response to the Covid-19. In terms of macroeconomic variables, the daily return of exchange rate and industrial production index reduced by 1.90% and 35.67%, respectively. The unemployment rate reached the all-time high with 5.30% whereas the interest rate has reduced to the lowest rate, that is 1.75%. Last but not least, the daily return of brent crude oil has reduced by 27.98% due to the lack of demand as the global economy was shut down because of pandemic.

Table 2 reports the correlation matrix of the data. It is acknowledged that the daily growth rate of Covid-19 has a negative (positive) correlation with stock return (volatility) and it appears that the variable is significant when determining the stock volatility instead of stock return. It is also acknowledged that the stringency index shows a positive correlation with stock return. From the non-pharmaceutical intervention (NPI) perspective, cancelling public events and international travel control have shown to have a positive correlation with stock return at 5% significant level. On the other hand, restrictions on gatherings, closing public transport as well as restrictions of internal movements have a significant and negative correlation with stock volatility during the covid period.

The baseline empirical findings are summarized in Table 3. In the estimated regression models, as summarized in Columns 1–3, the dependent variable is market return of FBMKLCI whereas Columns 4–6 use return volatility which is the natural logarithm of absolute return from FBMKLCI as dependent variable. Firstly, the study begins by scrutinizing the coefficient estimates summarized in columns 1–3. In the first column, the results show that growth rate of Covid-19 exerts a negative and significant effect (at 1% significance level) on stock return. A 1% increase in daily Covid-19 cases will reduce the FBMKLCI' returns by 0.026%. This finding is consistent with Ashraf [2] and Al-Awadhi et al. [1]. In the second column, it is also noticed that the Stringency Index enters a positive and significant at 1% level showing that the stock market reacts positively to the government's social distancing measure in containing the virus.

**Table 2. Correlation coefficients between variables.**

| | 1 | 2 | 3 | 4 | 5 | 6 | 7 | 8 | 9 | 10 | 11 | 12 | 13 | 14 | 15 | 16 | 17 | 18 |
|---|---|---|---|---|---|---|---|---|---|---|---|---|---|---|---|---|---|---|
| 1. $RET_{FBMKLCI}$ | 1 | | | | | | | | | | | | | | | | | |
| 2. $VOL_{FBMKLCI}$ | -0.061 | 1 | | | | | | | | | | | | | | | | |
| 3. CGRATE | -0.248** | 0.354** | 1 | | | | | | | | | | | | | | | |
| 4. SI | 0.119** | -0.001 | -0.190** | 1 | | | | | | | | | | | | | | |
| 5. C1 | 0.091* | 0.083* | -0.060 | 0.607** | 1 | | | | | | | | | | | | | |
| 6. C2 | 0.109** | 0.033 | -0.161** | 0.785** | 0.446** | 1 | | | | | | | | | | | | |
| 7. C3 | 0.116** | -0.037 | -0.267** | 0.677** | 0.440** | 0.572** | 1 | | | | | | | | | | | |
| 8. C4 | 0.047 | -0.114** | -0.213** | 0.510** | 0.252** | 0.250** | 0.586** | 1 | | | | | | | | | | |
| 9. C5 | -0.008 | -0.097* | -0.071 | 0.529** | 0.001 | 0.261** | 0.108** | 0.327** | 1 | | | | | | | | | |
| 10. C6 | 0.046 | -0.003 | -0.066 | 0.709** | 0.339** | 0.622** | 0.193** | 0.094* | 0.501** | 1 | | | | | | | | |
| 11. C7 | 0.114** | -0.118** | -0.240** | 0.654** | 0.364** | 0.444** | 0.635** | 0.512** | 0.247** | 0.418** | 1 | | | | | | | |
| 12. C8 | 0.066 | 0.191** | 0.030 | 0.418** | 0.271** | 0.360** | 0.066 | -0.044 | 0.208** | 0.393** | -0.111** | 1 | | | | | | |
| 13.INT | -0.061 | 0.214** | 0.344** | -0.497** | -0.323** | -0.361** | -0.651** | -0.631** | -0.335** | -0.114** | -0.518** | 0.020 | 1 | | | | | |
| 14. INF | -0.040 | -0.200** | -0.127* | -0.220** | -0.244** | -0.367** | 0.000 | 0.372** | 0.097* | -0.387** | 0.285** | -0.547** | -0.289** | 1 | | | | |
| 15. RIPI | -0.007 | 0.011 | -0.061 | -0.085* | 0.042 | -0.127** | 0.057 | 0.052 | 0.014 | -0.313** | -0.149** | 0.057 | -0.339** | 0.059 | 1 | | | |
| 16. REXRATE | 0.012 | -0.021 | -0.043 | -0.039 | -0.087* | -0.041 | -0.018 | -0.006 | -0.028 | -0.008 | 0.005 | -0.031 | 0.083* | 0.068 | -0.049 | 1 | | |
| 17. UNEMPLOY | 0.084* | -0.008 | -0.173** | 0.754** | 0.488** | 0.669** | 0.555** | 0.259** | 0.322** | 0.589** | 0.283** | 0.512** | -0.450** | -0.601** | 0.086* | -0.084* | 1 | |
| 18. Δ CRUDE OIL | 0.173** | 0.019 | -0.073 | 0.113** | 0.100* | 0.073 | 0.080 | 0.079 | 0.042 | 0.022 | 0.109** | 0.045 | -0.129** | -0.002 | 0.056 | 0.056 | 0.134** | 1 |

This table reports Pearson correlation coefficients between the different variables used in the study. VOL represents volatility which is the logarithm of absolute daily returns of FBMKLCI (Log| $RET_{t}$|). RET is daily market return of FBMKLCI. The daily growth rate of Covid-19 is expressed as CGRATE. SI is the Stringency Index which measured on a scale of 0 to 100 that represents the daily government response to COVID-19 by using non-pharmaceutical policies and eight sub-indices under this indicator are school closing (C1), workplace closing (C2), cancel public events (C3), restrictions on gathering (C4), close public transport (C5), stay-at-home requirement (C6), restrictions of internal movement (C7) and international travel controls (C8). CRUDE OIL is the daily price of brent crude oil. EXRATE is the daily exchange rate of USD/MYR. UNEMPLOY is the monthly rate of unemployment. IPI is the monthly index of industrial production. INT is the monthly overnight policy rate. INF is the monthly index of consumer price. The asterisks denote *, ** and *** statistical significance at 10%, 5% and 1% levels, respectively.

An increase of one index point of Stringency policy will increase the stock returns by 0.018%. This finding is in line with Phan & Narayan [27] and Wang et al. [16]. The aforementioned studies suggest that the implementation of social distancing measures may greatly damage stock market in the short term, but it can protect and recover the equity market given their contributions to mitigate the spread of Covid-19. Thus, subdue investor panic and strengthen investor confidence. However, this result contradicts Asraf [2], who find that stringent social distancing policy generates a significant and negative effect on the stock market return due to adverse impact on economic activity.

**Table 3. Main regression.**

| | Dependent variable: $FBMKLCI_{RET}$ | | | Dependent variable: $FBMKLCI_{VOL}$ | | |
|---|---|---|---|---|---|---|
| | **(1)** | **(2)** | **(3)** | **(4)** | **(5)** | **(6)** |
| CGRATE | -0.026*** | 0.002 | -0.026*** | 0.022*** | -0.005 | 0.022*** |
| | (0.000) | (0.901) | (0.001) | (0.001) | (0.672) | (0.001) |
| SI | 0.011 | 0.018*** | | 0.011** | 0.005 | |
| | (0.130) | (0.007) | | (0.029) | (0.256) | |
| *CGRATE×SI* | | -0.001** | | | 0.001** | |
| | | (0.051) | | | (0.025) | |
| C1 | | | 0.037 | | | 0.049 |
| | | | (0.427) | | | (0.132) |
| C2 | | | 0.043 | | | 0.050 |
| | | | (0.587) | | | (0.348) |
| C3 | | | 0.068 | | | 0.252** |
| | | | (0.689) | | | (0.038) |
| C4 | | | 0.007 | | | 0.021 |
| | | | (0.871) | | | (0.433) |
| C5 | | | 0.045 | | | 0.001 |
| | | | (0.594) | | | (0.989) |
| C6 | | | -0.081 | | | -0.010 |
| | | | (0.320) | | | (0.842) |
| C7 | | | 0.176 | | | -0.001 |
| | | | (0.113) | | | (0.986) |
| C8 | | | 0.106 | | | 0.178*** |
| | | | (0.206) | | | (0.004) |
| INT | -0.038 | -0.042 | 0.160 | 0.185 | 0.188 | 0.445* |
| | (0.894) | (0.884) | (0.643) | (0.310) | (0.302) | (0.059) |
| INF | -0.074* | -0.091** | -0.070 | -0.078*** | -0.061** | -0.046 |
| | (0.082) | (0.026) | (0.137) | (0.007) | (0.024) | (0.154) |
| RIPI | 0.001 | 0.001 | 0.000 | 0.008** | 0.008** | 0.007* |
| | (0.864) | (0.805) | (0.987) | (0.026) | (0.034) | (0.071) |
| REXRATE | -0.024 | -0.029 | -0.022 | -0.018 | -0.013 | -0.032 |
| | (0.862) | (0.839) | (0.870) | (0.859) | (0.889) | (0.749) |
| UNEMPLOY | -0.509 | -0.666** | -0.359 | -0.453* | -0.302 | -0.397* |
| | (0.157) | (0.051) | (0.279) | (0.079) | (0.195) | (0.081) |
| Δ CRUDE OIL | 0.044** | 0.044** | 0.042** | 0.010 | 0.010 | 0.011 |
| | (0.022) | (0.022) | (0.030) | (0.472) | (0.468) | (0.440) |
| Obs. | 575 | 575 | 575 | 575 | 575 | 575 |
| $R^2$ | 0.102 | 0.114 | 0.109 | 0.181 | 0.203 | 0.212 |
| Adjusted $R^2$ | 0.090 | 0.100 | 0.085 | 0.170 | 0.190 | 0.190 |

(*Continued*)

**Table 3.** (Continued)

| | Dependent variable: $FBMKLCI_{RET}$ | | | Dependent variable: $FBMKLCI_{VOL}$ | | |
|---|---|---|---|---|---|---|
| | **(1)** | **(2)** | **(3)** | **(4)** | **(5)** | **(6)** |
| F-stat | 8.094*** | 8,115*** | 4.550*** | 15.671*** | 16.023*** | 10.015*** |

This table presents the results of the OLS regression with bootstrapping technique based on 10,000 bootstrap sample regarding the impact of non-pharmaceutical policies in controlling COVID-19 pandemic on stock market returns and volatility. The dependent variable are FBMKLCI's return or volatility which expressed in *RET* and *VOL* in logarithm of absolute daily returns of FBMKLCI ($Log|RET|_t$) in Column 1–3 and Column 4–6, respectively. The independent variables are daily growth rate of COVID-19 (CGRATE), the Stringency Index (SI) and the different type of non-pharmaceutical interventions implemented in the country including school closing (C1), workplace closing (C2), cancel public events (C3), restrictions on gathering (C4), close public transport (C5), stay-at-home requirement (C6), restrictions of internal movement (C7) and international travel controls (C8). Also, a set of control variables including Δ CRUDE OIL is the change in daily price of brent crude oil, EXRATE is the daily return of exchange rate of USD/MYR, UNEMPLOY is the monthly rate of unemployment, RIPI is the rate of monthly index of industrial production, INT is the monthly overnight policy rate and INF is the monthly index of consumer price. The number in the brackets are p-value and asterisks denote *, ** and *** statistical significance at 10%, 5% and 1% levels, respectively.

Following that, the model estimates how government actions interact with the growth in Covid-19 confirmed cases to affect stock market returns by implementing indirect effect of non-pharmaceutical intervention variable. As depicted in Column 2 of Table 3, the interaction term, *Stringency Index × CGRATE*, enters a negative and significant effect (at 5% significant level) on stock market returns. A possible explanation is that it could be due to inconsistent levels of responses by the government in dealing with the virus contamination that contributes to accelerating of daily confirmed cases in Malaysia. The increase subsequently led to unending lockdown and the closure of business. This finding is contradicted to the study of Ashraf [2] which proposed that stringent government social distancing measures are likely to weaken the stock markets' negative reaction to the growth of Covid-19 confirmed cases. To further investigate which individual government policies have significant effect on the stock market returns, this study has summarized the eight different types of social distancing measure in response to the Covid-19 outbreak as illustrated in Column 3. It can be observed that none of these proposed social distancing policies' responses display a significant regression coefficient when rate return of stock market is used as dependent variable in the model specification. Thus, these results suggest investors expect the social distancing measures are less effective mechanism to contain the disease.

Next, Column 4–6 in Table 3 illustrates the imposition of non-pharmaceutical policies on market return volatility. Consistent with the finding of Zaremba et al. [10] and Baig et al. [14], the growth rate of Covid-19 has positive and significant effect (at 1% significant level) on returns volatility. This indicates that an additional one percent increase in growth rate of Covid-19 will increase the return volatility by 0.005% to 0.022% as illustrated in Column 4 and 6. In addition to this, the estimated coefficient of Stringency index shows a positive and statistically significant at 5% level. This means that an increase of one index point of stringency policy will trigger the growth of volatility by 0.011%. A possible explanation is that it could be due to constant flow of policy-related news such as inconsistent government policies which may lead to potential divergence of opinions and eventually, increase in trading activity and elevates the growth of volatility (Banerjee [28]; Foucault, Sraer and Thesmar [29]; Harris and Raviv [30]; and Manela and Moreira [31]). This can be witnessed from unclear communication on the enforcement of SOPs for businesses.

Another important finding is the interaction term, *Stringency Index × CGRATE* also enters a positive and statistically significant at 5% level in determining the growth of volatility. In column 6, the study reveals the implication from using non-pharmaceutical intervention (NPI) by government in response to COVID-19 on stock return volatility. Of all the eight NPIs introduced in the country, closing public events and international travel control are the only social distancing measures that has positive and significant effect on the stock return volatility at least at 5% significant level. A possible explanation is that it could be the closure of the international border which led to a disruption in foreign and domestic tourism economy especially for travel industry. In addition to this, closing public events will disrupt local businesses which eventually affect domestic economy as well. Hence, an increase in restriction level in international travel (cancelling public events) triggers the growth of return volatility by 0.18% (0.25%) as the investors were unable to predict the reopening of the border as well as the resumption of economic activity due to uncertainty caused by the deadly virus in the near future. Apart from shutting down the international border, social distancing policies such as stay-at-home (C6) and restriction on internal movement (C7) policies showed a reduction in the growth of returns volatility although they are less effective mechanisms to contain the virus and perhaps testing and quarantine policies should be tested for future studies.

As for control variable, inflation rate has shown negative and significant effect across all the model specification except when individual social distancing policy is incorporated as independent variable. Using stock market return as dependent variable, the estimated coefficient of inflation rate in column 1 and 2 are -0.074% and -0.091%, respectively with at least 10% significant level. On the contrary, the estimated coefficient of inflation rate ranges from -0.061% to -0.078% and significant at least at 5% when return volatility is implemented as dependent variable. In column 2, the unemployment rate has a negative and significant on the stock market return. The estimated coefficient of unemployment rate is -0.666% which means that one percent increase in unemployment rate will reduce the stock market return by -0.666%. In column 4, the unemployment rate has a negative and statistically significant at 10% level on return volatility. This means one percent increase in unemployment rate will reduce the growth of volatility by 0.397%-0.453%. For rate return of industrial production index, it has positive and significant relationship with return volatility. The estimated coefficient of RIPI can be ranged from 0.007% to 0.008% with at least 10% significant level. This indicates that one percent increase in RIPI triggers an additional growth in the volatility by 0.007% to 0.008% in the mid COVID period.

## Robustness checks

This paper also undertakes a number of robustness checks to ensure validity of results. The robustness checks are displayed in Tables 4 to 11. Table 4–6 summarize the estimation results from regression models with alternative measures of the dependent variables from different sector indices return. It is evident that the estimated coefficient of CGRATE exerts negative and significant effect on all sector indices returns at least at 5% significant level except plantation and transport and logistic sector with no significant change. From the Stringency index variable, 15 out of 24 coefficients of *SI* are positive and significant at the 10% level at least, that is, government responses to Covid-19 are capable of reducing the negative shock on the stock return of different sectors. For 7 out of 12 sectors, the variable of the interaction term (*CGRATE*SI*) exhibits negative and significant at least at 10% level. Apart from that, it can be observed the non-pharmaceutical policies has diverse impact on stock returns of different sectors. For instance, 5 out of 12 sectors (namely construction, financial service, industrial product and service, property and transportation and logistics sector) react positive and significant

**Table 4. Robustness tests: Alternative dependent variables using different sector indices returns.**

| | $RET_{CONST}$ | | | $RET_{FIN}$ | | | $RET_{IND}$ | | | $RET_{PLANT}$ | | |
|---|---|---|---|---|---|---|---|---|---|---|---|---|
| | (1) | (2) | (3) | (4) | (5) | (6) | (7) | (8) | (9) | (10) | (11) | (12) |
| CGRATE | -0.060*** | -0.011 | -0.059*** | -0.037*** | 0.015 | -0.036*** | -0.036*** | 0.02 | -0.035** | -0.009 | 0.009 | -0.009 |
| | (0.000) | (0.679) | (0.000) | (0.000) | (0.295) | (0.000) | (0.009) | (0.938) | (0.012) | (0.292) | (0.665) | (0.282) |
| SI | 0.016 | 0.027*** | | 0.020** | 0.032*** | | 0.015 | 0.024** | | 0.012 | 0.016** | |
| | (0.151) | (0.008) | | (0.045) | (0.001) | | (0.171) | (0.016) | | (0.167) | (0.042) | |
| CGRATE*SI | | -0.001* | | | -0.001*** | | | -0.001 | | | 0.000 | |
| | | (0.072) | | | (0.005) | | | (0.121) | | | (0.371) | |
| C1 | | | 0.025 | | | 0.041 | | | 0.063 | | | 0.125** |
| | | | (0.719) | | | (0.429) | | | (0.359) | | | (0.037) |
| C2 | | | 0.218* | | | 0.117 | | | 0.154 | | | 0.099 |
| | | | (0.058) | | | (0.204) | | | (0.178) | | | (0.377) |
| C3 | | | -0.014 | | | -0.093 | | | -0.152 | | | 0.202 |
| | | | (0.961) | | | (0.678) | | | (0.560) | | | (0.353) |
| C4 | | | 0.006 | | | 0.030 | | | -0.003 | | | 0.025 |
| | | | (0.906) | | | (0.562) | | | (0.949) | | | (0.630) |
| C5 | | | 0.069 | | | 0.112 | | | 0.027 | | | 0.071 |
| | | | (0.574) | | | (0.254) | | | (0.816) | | | (0.537) |
| C6 | | | -0.113 | | | -0.088 | | | -0.131 | | | -0.115 |
| | | | (0.350) | | | (0.337) | | | (0.284) | | | (0.337) |
| C7 | | | 0.344* | | | 0.285** | | | 0.326* | | | 0.051 |
| | | | (0.049) | | | (0.038) | | | (0.058) | | | (0.681) |
| C8 | | | 0.025 | | | 0.191 | | | 0.114 | | | 0.035 |
| | | | (0.850) | | | (0.105) | | | (0.364) | | | (0.736) |
| Control var. | Yes | Yes | Yes | Yes | Yes | Yes | Yes | Yes | Yes | Yes | Yes | Yes |
| Obs. | 575 | 575 | 575 | 575 | 575 | 575 | 575 | 575 | 575 | 575 | 575 | 575 |
| $R^2$ | 0.179 | 0.193 | 0.193 | 0.138 | 0.163 | 0.143 | 0.135 | 0.145 | 0.144 | 0.049 | 0.052 | 0.064 |
| Adjusted $R^2$ | 0.167 | 0.180 | 0.172 | 0.126 | 0.150 | 0.120 | 0.123 | 0.131 | 0.121 | 0.036 | 0.037 | 0.039 |
| F-stat | 15.445*** | 15.029*** | 8.957*** | 11.328*** | 12.269*** | 6.219*** | 11.056*** | 10.654*** | 6.297*** | 3.678*** | 3.454*** | 2.537*** |

This table presents the results of different robustness test. In the regression of Table 5, alternative sector indices' returns as dependent variables are used, namely, returns of construction index ($RET_{CONS}$), financial index ($RET_{FIN}$), industrial product and services ($RET_{IND}$), plantation ($RET_{PLANT}$). The independent variables are daily growth rate of COVID-19 (CGRATE), the Stringency Index (SI) and the different type of non-pharmaceutical interventions implemented in the country including school closing (C1), workplace closing (C2), cancel public events (C3), restrictions on gathering (C4), close public transport (C5), stay-at-home requirement (C6), restrictions of internal movement (C7) and international travel controls (C8). The number in the brackets are p-value and asterisks denote *, ** and *** statistical significance at 10%, 5% and 1% levels, respectively.

towards restriction of internal movement (C7) policy. A possible explanation could be, the implementation of such policy at an early stage of pandemic mitigates the virus within the country from wide spread which eventually reduce the disruption in economic activity and promotes positive return. Tables 7 to 9 show the results of the time series model using stock volatility as dependent variable of different sectors. Twenty-one out of thirty-six coefficient of CGRATE are positive and significant at the 10% at least, that is, every one percent increase in daily growth rate of Covid-19 cases will contribute to the growth of volatility of the affected sectors. From Stringency index variable, 14 out of 24 coefficients of $SI$ are positive and significant at least at 10% level. For 8 out of 12 sectors, the variable of interaction terms ($CGRATE^*SI$) possesses significant and positive effect on return volatility of the sectors. Following the results

**Table 5. Robustness tests: Alternative dependent variables using different sector indices returns.**

| | $RET_{PROPERTY}$ | | | $RET_{CON}$ | | | $RET_{TECH}$ | | | $RET_{ENERGY}$ | | |
|---|---|---|---|---|---|---|---|---|---|---|---|---|
| | (13) | (14) | (15) | (16) | (17) | (18) | (19) | (20) | (21) | (22) | (23) | (24) |
| CGRATE | -0.038*** | 0.013 | -0.038*** | -0.030*** | 0.004 | -0.029*** | -0.061*** | 0.008 | -0.060*** | -0.085** | -0.033 | -0.087*** |
| | (0.000) | (0.445) | (0.000) | (0.000) | (0.786) | (0.001) | (0.004) | (0.859) | (0.003) | (0.006) | (0.659) | (0.006) |
| SI | 0.017** | 0.029*** | | 0.016*** | 0.024*** | | 0.006 | 0.022 | | 0.036* | 0.048*** | |
| | (0.046) | (0.000) | | (0.009) | (0.000) | | (0.723) | (0.170) | | (0.061) | (0.004) | |
| CGRATE*SI | | -0.001*** | | | -0.001** | | | -0.002* | | | -0.001 | |
| | | (0.008) | | | (0.018) | | | (0.074) | | | (0.304) | |
| C1 | | | 0.032 | | | 0.075* | | | 0.017 | | | 0.222* |
| | | | (0.563) | | | (0.075) | | | (0.868) | | | (0.078) |
| C2 | | | 0.145 | | | 0.159** | | | 0.251 | | | 0.207 |
| | | | (0.118) | | | (0.014) | | | (0.145) | | | (0.323) |
| C3 | | | -0.033 | | | 0.007 | | | -0.070 | | | 0.110 |
| | | | (0.877) | | | (0.958) | | | (0.833) | | | (0.806) |
| C4 | | | 0.020 | | | 0.005 | | | -0.077 | | | 0.106 |
| | | | (0.675) | | | (0.876) | | | (0.249) | | | (0.330) |
| C5 | | | 0.069 | | | 0.060 | | | -0.022 | | | 0.334 |
| | | | (0.487) | | | (0.400) | | | (0.897) | | | (0.116) |
| C6 | | | 0.008 | | | -0.040 | | | 0.109 | | | -0.038 |
| | | | (0.936) | | | (0.578) | | | (0.528) | | | (0.853) |
| C7 | | | 0.248* | | | 0.121 | | | -0.155 | | | 0.270 |
| | | | (0.059) | | | (0.228) | | | (0.462) | | | (0.331) |
| C8 | | | 0.054 | | | 0.094 | | | -0.176 | | | 0.009 |
| | | | (0.616) | | | (0.264) | | | (0.336) | | | (0.973) |
| Control var. | Yes | Yes | Yes | Yes | Yes | Yes | Yes | Yes | Yes | Yes | Yes | Yes |
| Obs. | 575 | 575 | 575 | 575 | 575 | 575 | 575 | 575 | 575 | 575 | 575 | 575 |
| $R^2$ | 0.175 | 0.200 | 0.185 | 0.171 | 0.190 | 0.180 | 0.103 | 0.118 | 0.112 | 0.199 | 0.204 | 0.205 |
| Adjusted $R^2$ | 0.163 | 0.168 | 0.163 | 0.159 | 0.177 | 0.158 | 0.091 | 0.104 | 0.088 | 0.188 | 0.191 | 0.184 |
| F-stat | 15.046*** | 15.765*** | 8.484*** | 14.589*** | 14.771*** | 8.214*** | 8.166*** | 8.386*** | 4.702*** | 17.596*** | 16.129*** | 9.654*** |

This table presents the results of different robustness test. In the regression of Table 6, alternative sector indices' returns as dependent variables are used, namely, returns of property index ($RET_{PROPERTY}$), consumer and product services ($RET_{CON}$), technology ($RET_{TECH}$), energy ($RET_{ENERGY}$). The independent variables are daily growth rate of COVID-19 (CGRATE), the Stringency Index (SI) and the different type of non-pharmaceutical interventions implemented in the country including school closing (C1), workplace closing (C2), cancel public events (C3), restrictions on gathering (C4), close public transport (C5), stay-at-home requirement (C6), restrictions of internal movement (C7) and international travel controls (C8). The number in the brackets are p-value and asterisks denote *, ** and *** statistical significance at 10%, 5% and 1% levels, respectively.

from the baseline findings in Table 3, there are two non-pharmaceutical policies remain intact: cancel public events (C3) and international travel restrictions (C8) policies. Such policies have positive and significant effect on return volatility for at least more than half of the sectors.

The robustness checks in Tables 10 and 11 concentrate on additional control variables that might affect the stock market return and volatility. Among the additional control variables are daily death growth rate of Covid-19, Log (daily vaccination) and weekday dummies) which tested for seasonal effect. First, this study incorporates daily death growth rate of Covid-19 to the empirical model and compares the main findings with the former analysis. Previous studies suggest that daily growth rate of death caused by Covid-19 has negative effect on stock market returns (Asraf [2]; Dharani et al. [32]) and positive effect on stock volatility (Baek & Lee [33]). Panel A of Table 10 presents the corresponding results: Majority of the main findings

**Table 6. Robustness tests: Alternative dependent variables using different sector indices returns.**

| | $RET_{HEALTH}$ | | | $RET_{TELCO}$ | | | $RET_{TRANSPORT}$ | | | $RET_{UTILITIES}$ | | |
|---|---|---|---|---|---|---|---|---|---|---|---|---|
| | (25) | (26) | (27) | (28) | (29) | (30) | (31) | (32) | (33) | (34) | (35) | (36) |
| CGRATE | -0.040*** | -0.064*** | -0.039*** | -0.025*** | 0.019 | -0.024*** | -0.028 | -0.002 | -0.028** | -0.031*** | 0.002 | -0.030*** |
| | (0.000) | (0.008) | (0.000) | (0.002) | (0.194) | (0.003) | (0.673) | (0.943) | (0.012) | (0.001) | (0.902) | (0.002) |
| SI | -0.021* | -0.027** | | 0.014* | 0.024*** | | 0.009 | 0.015 | | 0.011* | 0.019** | |
| | (0.051) | (0.030) | | (0.084) | (0.002) | | (0.391) | (0.108) | | (0.098) | (0.005) | |
| CGRATE*SI | | 0.001 | | | -0.001*** | | | -0.001 | | | -0.001* | |
| | | (0.230) | | | (0.004) | | | (0.302) | | | (0.053) | |
| C1 | | | -0.059 | | | 0.028 | | | 0.070 | | | 0.019 |
| | | | (0.587) | | | (0.631) | | | (0.280) | | | (0.656) |
| C2 | | | -0.120 | | | 0.163* | | | -0.012 | | | 0.132* |
| | | | (0.405) | | | (0.067) | | | (0.906) | | | (0.055) |
| C3 | | | 0.166 | | | -0.046 | | | 0.067 | | | 0.034 |
| | | | (0.613) | | | (0.813) | | | (0.776) | | | (0.817) |
| C4 | | | 0.018 | | | 0.030 | | | -0.015 | | | -0.008 |
| | | | (0.824) | | | (0.409) | | | (0.763) | | | (0.790) |
| C5 | | | -0.082 | | | -0.001 | | | 0.131 | | | -0.001 |
| | | | (0.678) | | | (0.995) | | | (0.276) | | | (0.985) |
| C6 | | | -0.391* | | | 0.020 | | | -0.074 | | | -0.043 |
| | | | (0.060) | | | (0.854) | | | (0.478) | | | (0.585) |
| C7 | | | 0.110 | | | 0.169 | | | 0.266* | | | 0.149 |
| | | | (0.730) | | | (0.315) | | | (0.098) | | | (0.145) |
| C8 | | | -0.317* | | | 0.077 | | | -0.048 | | | 0.021 |
| | | | (0.069) | | | (0.459) | | | (0.693) | | | (0.791) |
| Control var. | Yes | Yes | Yes | Yes | Yes | Yes | Yes | Yes | Yes | Yes | Yes | Yes |
| Obs. | 575 | 575 | 575 | 575 | 575 | 575 | 575 | 575 | 575 | 575 | 575 | 575 |
| $R^2$ | 0.0608 | 0.070 | 0.097 | 0.098 | 0.116 | 0.108 | 0.093 | 0.098 | 0.104 | 0.150 | 0.169 | 0.160 |
| Adjusted $R^2$ | 0.055 | 0.055 | 0.73 | 0.086 | 0.102 | 0.084 | 0.080 | 0.084 | 0.080 | 0.138 | 0.156 | 0.137 |
| F-stat | 5.197*** | 4.754*** | 4.003*** | 7.728*** | 8.266*** | 4.529*** | 7.268*** | 6.856*** | 4.318*** | 12.553*** | 12.778*** | 7.107*** |

This table presents the results of different robustness test. In the regression of Table 7, alternative sector indices' returns as dependent variables are used, namely, returns of healthcare ($RET_{HEALTH}$), telecommunication and media index ($RET_{TELCO}$), transportation and logistic index ($RET_{TRANSPORT}$), utilities index ($RET_{UTILITIES}$ The independent variables are daily growth rate of COVID-19 (CGRATE), the Stringency Index (SI) and the different type of non-pharmaceutical interventions implemented in the country including school closing (C1), workplace closing (C2), cancel public events (C3), restrictions on gathering (C4), close public transport (C5), stay-at-home requirement (C6), restrictions of internal movement (C7) and international travel controls (C8). The number in the brackets are p-value and asterisks denote *, ** and *** statistical significance at 10%, 5% and 1% levels, respectively.

remain intact after the inclusion of daily death growth rate of Covid-19. The stringency index exhibits a positive and significant effect on stock return (volatility) in column 2 (column 4). However, the variable of the interaction term (CGRATE*SI) shows insignificant effect on stock market return in column 2. On the positive note, there is a significant and positive relationship between social distancing policy and stock return volatility particularly for public cancel events (C3) and international travel restrictions (C8) as depicted in column 6. Second, this study employs log (daily vaccination) to both market features (return and volatility) to ensure the main findings remain robust. The literature shows that Covid-19 vaccine plays a significant role in determining stock market returns as well as volatility. It is said that Covid-19 vaccination leads to a rise in mean stock returns (Apergis, Mustafa & Malik [34]) and has a stronger

**Table 7. Robustness tests: Alternative dependent variables using different sector indices volatility.**

| | $VOL_{CONST}$ | | | $VOL_{FIN}$ | | | $VOL_{IND}$ | | | $VOL_{PLANT}$ | | |
|---|---|---|---|---|---|---|---|---|---|---|---|---|
| | (1) | (2) | (3) | (4) | (5) | (6) | (7) | (8) | (9) | (10) | (11) | (12) |
| CGRATE | 0.054*** | -0.007 | 0.055*** | 0.032*** | -0.007 | 0.031*** | 0.031** | 0.006 | 0.031** | 0.003 | -0.018 | 0.004 |
| | (0.000) | (0.774) | (0.000) | (0.000) | (0.633) | (0.000) | (0.018) | (0.842) | (0.016) | (0.699) | (0.307) | (0.643) |
| SI | 0.015** | 0.000 | | 0.023*** | 0.014** | | 0.017** | 0.012* | | 0.009 | 0.004 | |
| | (0.034) | (0.971) | | (0.002) | (0.038) | | (0.030) | (0.094) | | (0.184) | (0.501) | |
| CGRATE*SI | | 0.002*** | | | 0.001*** | | | 0.001 | | | 0.001 | |
| | | (0.003) | | | (0.008) | | | (0.297) | | | (0.154) | |
| C1 | | | 0.004 | | | 0.137*** | | | 0.060 | | | 0.043 |
| | | | (0.927) | | | (0.000) | | | (0.217) | | | (0.308) |
| C2 | | | 0.065 | | | 0.148** | | | 0.104 | | | 0.067 |
| | | | (0.405) | | | (0.027) | | | (0.197) | | | (0.427) |
| C3 | | | 0.360* | | | 0.158 | | | 0.505*** | | | 0.420*** |
| | | | (0.045) | | | (0.334) | | | (0.007) | | | (0.009) |
| C4 | | | 0.033 | | | 0.090** | | | 0.006 | | | -0.027 |
| | | | (0.373) | | | (0.017) | | | (0.868) | | | (0.515) |
| C5 | | | 0.040 | | | 0.037 | | | 0.067 | | | 0.071 |
| | | | (0.612) | | | (0.618) | | | (0.365) | | | (0.412) |
| C6 | | | -0.066 | | | -0.035 | | | -0.144* | | | -0.090 |
| | | | (0.401) | | | (0.569) | | | (0.070) | | | (0.250) |
| C7 | | | 0.051 | | | 0.051 | | | 0.066 | | | 0.108 |
| | | | (0.677) | | | (0.605) | | | (0.556) | | | (0.205) |
| C8 | | | 0.252** | | | 0.223** | | | 0.257** | | | -0.063 |
| | | | (0.025) | | | (0.021) | | | (0.010) | | | (0.456) |
| Control var. | Yes | Yes | Yes | Yes | Yes | Yes | Yes | Yes | Yes | Yes | Yes | Yes |
| Obs. | 575 | 575 | 575 | 575 | 575 | 575 | 575 | 575 | 575 | 575 | 575 | 575 |
| $R^2$ | 0.301 | 0.339 | 0.320 | 0.222 | 0.245 | 0.251 | 0.220 | 0.227 | 0.266 | 0.053 | 0.060 | 0.097 |
| Adjusted $R^2$ | 0.291 | 0.328 | 0.302 | 0.211 | 0.233 | 0.231 | 0.220 | 0.215 | 0.246 | 0.040 | 0.046 | 0.072 |
| F-stat | 30.55*** | 32.223*** | 17.597*** | 20.212*** | 20.399*** | 12.504*** | 20.004*** | 18.501*** | 13.508*** | 3.971*** | 4.048*** | 3.992*** |

This table presents the results of different robustness test. In the regression of Table 8, alternative sector indices' volatility as dependent variables are used, namely, returns volatility of construction index ($VOL_{CONST}$, $|\log|RET_{CONST}||$), financial index ($VOL_{FIN}$, $|\log|RET_{FIN}||$), industrial product and services ($VOL_{IND}$, $|\log|RET_{IND}||$), plantation ($VOL_{PLANT}$, $|\log|RET_{PLANT}||$). The independent variables are daily growth rate of COVID-19 (CGRATE), the Stringency Index (SI) and the different type of non-pharmaceutical interventions implemented in the country including school closing (C1), workplace closing (C2), cancel public events (C3), restrictions on gathering (C4), close public transport (C5), stay-at-home requirement (C6), restrictions of internal movement (C7) and international travel controls (C8). The number in the brackets are p-value and asterisks denote *, ** and *** statistical significance at 10%, 5% and 1% levels, respectively.

effect in reducing stock market volatility (Routbi et al. [35]). Panel B of Table 10 presents the corresponding results: All results are similar to the baseline results in Table 3. In particular, Stringency index yield a significant positive effect on stock market returns (volatility) in Column 2 (Column 4). In addition to this, the moderation effect from the interaction terms lead to significant negative (positive) effect on stock market returns (volatility) as illustrated in Column 2 (Column 5) of Table 10. In terms of social distancing policy, international travel controls (C8) remain intact and still a major contributor to the growth of stock volatility whereas cancel public events (C3) does not lead to insignificant positive effect on stock volatility in column 6 as illustrated in Panel B of Table 10. Finally, this study employs weekday dummies into all the empirical model on both market features (return and volatility) in order to examine for

**Table 8. Robustness tests: Alternative dependent variables using different sector indices volatility.**

| | $VOL_{PROPERTY}$ | | | $VOL_{CONS}$ | | | $VOL_{TECH}$ | | | $VOL_{ENERGY}$ | | |
|---|---|---|---|---|---|---|---|---|---|---|---|---|
| | (13) | (14) | (15) | (16) | (17) | (18) | (19) | (20) | (21) | (22) | (23) | (24) |
| CGRATE | 0.035*** | -0.013 | 0.035*** | 0.025*** | 0.000 | 0.025*** | 0.052*** | 0.002 | 0.054*** | 0.071** | 0.017 | 0.071** |
| | (0.000) | (0.320) | (0.000) | (0.000) | (0.977) | (0.000) | (0.000) | (0.938) | (0.000) | (0.013) | (0.837) | (0.014) |
| SI | 0.019*** | 0.008* | | 0.014*** | 0.008* | | 0.033*** | 0.022** | | 0.029* | 0.017 | |
| | (0.000) | (0.072) | | (0.002) | (0.049) | | (0.002) | (0.013) | | (0.060) | (0.191) | |
| CGRATE*SI | | 0.001*** | | | 0.001** | | | 0.001** | | | 0.001 | |
| | | (0.001) | | | (0.030) | | | (0.034) | | | (0.265) | |
| C1 | | | 0.085** | | | 0.028 | | | 0.140** | | | 0.103 |
| | | | (0.027) | | | (0.331) | | | (0.038) | | | (0.326) |
| C2 | | | 0.030 | | | 0.089* | | | 0.603*** | | | 0.106 |
| | | | (0.632) | | | (0.047) | | | (0.000) | | | (0.544) |
| C3 | | | 0.275* | | | 0.269*** | | | 0.639*** | | | 0.972** |
| | | | (0.061) | | | (0.007) | | | (0.004) | | | (0.012) |
| C4 | | | 0.083** | | | 0.046** | | | 0.046 | | | -0.030 |
| | | | (0.018) | | | (0.030) | | | (0.324) | | | (0.744) |
| C5 | | | 0.011 | | | 0.019 | | | -0.108 | | | 0.237 |
| | | | (0.859) | | | (0.698) | | | (0.348) | | | (0.146) |
| C6 | | | 0.072 | | | -0.024 | | | -0.105 | | | 0.068 |
| | | | (0.202) | | | (0.602) | | | (0.356) | | | (0.654) |
| C7 | | | 0.030 | | | 0.018 | | | -0.238 | | | -0.046 |
| | | | (0.746) | | | (0.780) | | | (0.110) | | | (0.833) |
| C8 | | | 0.177* | | | 0.153** | | | 0.098 | | | 0.242 |
| | | | (0.080) | | | (0.016) | | | (0.391) | | | (0.252) |
| Control var. | Yes | Yes | Yes | Yes | Yes | Yes | Yes | Yes | Yes | Yes | Yes | Yes |
| Obs. | 575 | 575 | 575 | 575 | 575 | 575 | 575 | 575 | 575 | 575 | 575 | 575 |
| $R^2$ | 0.274 | 0.313 | 0.295 | 0.262 | 0.281 | 0.297 | 0.211 | 0.226 | 0.290 | 0.201 | 0.210 | 0.220 |
| Adjusted $R^2$ | 0.263 | 0.302 | 0.276 | 0.251 | 0.270 | 0.278 | 0.200 | 0.214 | 0.271 | 0.190 | 0.197 | 0.199 |
| F-stat | 26.682*** | 28.659*** | 15.600*** | 25.104*** | 24.637*** | 15.792*** | 18.966*** | 18.347*** | 15.259*** | 17.807*** | 16.700*** | 10.513*** |

This table presents the results of different robustness test. In the regression of Table 9, alternative sector indices' volatility as dependent variables are used, namely, returns volatility of property index ($VOL_{PROPERTY}$, $|\log|RET_{PROPERTY}||$), consumer and product services index ($VOL_{CONS}$, $|\log|RET_{CONS}||$), technology ($VOL_{TECH}$, $|\log|RET_{TECH}||$), energy ($VOL_{ENERGY}$, $|\log|RET_{ENERGY}||$). The independent variables are daily growth rate of COVID-19 (CGRATE), the Stringency Index (SI) and the different type of non-pharmaceutical interventions implemented in the country including school closing (C1), workplace closing (C2), cancel public events (C3), restrictions on gathering (C4), close public transport (C5), stay-at-home requirement (C6), restrictions of internal movement (C7) and international travel controls (C8). The number in the brackets are p-value and asterisks denote *, ** and *** statistical significance at 10%, 5% and 1% levels, respectively.

seasonal effect (Routbi et al. [35]; Liew et al. [36]). Based on Panel C in Table 11, the results suggest that all the main findings are similar to the former analysis. However, the interaction effect of social distancing policy and daily growth rate in cases caused by Covid-19 exhibits insignificant negative effect on stock market returns as depicted in column 2.

## Conclusion

The takeaway is clear: There is a trade-off between saving the live and saving the economy when non-pharmaceutical policies were implemented during the pandemic. One of the main takeaways from the findings is that when the non-pharmaceutical policies were not done properly, not only the daily Covid-19 confirmed cases increased but it also did more harm to the

**Table 9. Robustness tests: Alternative dependent variables using different sector indices volatility.**

| | $VOL_{HEALTH}$ | | | $VOL_{TELCO}$ | | | $VOL_{TRANSPORT}$ | | | $VOL_{UTILITIES}$ | | |
|---|---|---|---|---|---|---|---|---|---|---|---|---|
| | (25) | (26) | (27) | (28) | (29) | (30) | (31) | (32) | (33) | (34) | (35) | (36) |
| CGRATE | 0.017 | -0.002 | 0.021** | 0.017*** | -0.004 | 0.017*** | 0.029*** | -0.019 | 0.027* | 0.029*** | -0.003 | 0.029*** |
| | (0.109) | 0.885 | (0.011) | (0.001) | (0.686) | (0.000) | (0.008) | (0.287) | (0.010) | (0.000) | (0.838) | (0.000) |
| SI | -0.012* | -0.017** | | 0.014** | 0.009* | | 0.014* | 0.003 | | 0.013*** | 0.005 | |
| | (0.071) | (0.034) | | (0.011) | (0.075) | | (0.061) | (0.626) | | (0.005) | (0.212) | |
| CGRATE*SI | | 0.001 | | | 0.001** | | | 0.001** | | | 0.001** | |
| | | (0.168) | | | (0.029) | | | (0.01) | | | (0.016) | |
| C1 | | | -0.162** | | | 0.049 | | | 0.160*** | | | 0.036 |
| | | | (0.018) | | | (0.212) | | | (0.000) | | | (0.213) |
| C2 | | | -0.092 | | | 0.053 | | | 0.047 | | | 0.120** |
| | | | (0.346) | | | (0.357) | | | (0.502) | | | (0.011) |
| C3 | | | 0.363 | | | 0.427** | | | 0.196 | | | 0.238** |
| | | | (0.121) | | | (0.001) | | | (0.252) | | | (0.028) |
| C4 | | | -0.013 | | | 0.047* | | | 0.086** | | | 0.018 |
| | | | (0.819) | | | (0.072) | | | (0.010) | | | (0.427) |
| C5 | | | -0.475*** | | | 0.008 | | | 0.101 | | | -0.047 |
| | | | (0.000) | | | (0.910) | | | (0.223) | | | (0.323) |
| C6 | | | 0.017 | | | -0.071 | | | -0.141* | | | -0.022 |
| | | | (0.906) | | | (0.274) | | | (0.068) | | | (0.679) |
| C7 | | | -0.046 | | | 0.012 | | | 0.005 | | | 0.029 |
| | | | (0.835) | | | (0.916) | | | (0.958) | | | (0.699) |
| C8 | | | -0.026 | | | 0.091 | | | 0.186* | | | 0.079 |
| | | | (0.844) | | | (0.154) | | | (0.033) | | | (0.230) |
| Control var. | Yes | Yes | Yes | Yes | Yes | Yes | Yes | Yes | Yes | Yes | Yes | Yes |
| Obs. | 575 | 575 | 575 | 575 | 575 | 575 | 575 | 575 | 575 | 575 | 575 | 575 |
| $R^2$ | 0.166 | 0.168 | 0.204 | 0.160 | 0.168 | 0.201 | 0.207 | 0.242 | 0.251 | 0.282 | 0.313 | 0.315 |
| Adjusted $R^2$ | 0.154 | 0.155 | 0.182 | 0.148 | 0.155 | 0.180 | 0.196 | 0.230 | 0.231 | 0.272 | 0.302 | 0.297 |
| F-stat | 14.076*** | 12.701*** | 9.542*** | 13.480*** | 12.701*** | 9.400*** | 18.521*** | 20.075*** | 12.521*** | 27.841 | 28.648*** | 17.199*** |

This table presents the results of different robustness test. In the regression of Table 10, alternative sector indices' returns volatility as dependent variables are used, namely returns volatility of Healthcare index ($VOL_{HEALTHCARE}$, $|\log|RET_{HEALTHCARE}||$), Telcommunication and media ($VOL_{TELCO}$, $|\log|RET_{TELCO}||$), Transportation and Logistics ($VOL_{TRANSPORT}$, $|\log|RET_{TRANSPORT}||$), Utilities ($VOL_{utilities}$, $|\log|RET_{utilities}||$). The independent variables are daily growth rate of COVID-19 (CGRATE), the Stringency Index (SI) and the different type of non-pharmaceutical interventions implemented in the country including school closing (C1), workplace closing (C2), cancel public events (C3), restrictions on gathering (C4), close public transport (C5), stay-at-home requirement (C6), restrictions of internal movement (C7) and international travel controls (C8). The number in the brackets are p-value and asterisks denote *, ** and *** statistical significance at 10%, 5% and 1% levels, respectively.

equity market in Malaysia. Evidence found shows that the announcement regarding the social distancing measures taken by government in Malaysia's context were counterproductive as opposed to the previous study done by Asraf [2]. For instance, the Stringency index which is meant to limit Covid-19 infection rate has a positive effect on the stock market. This is because the country had imposed social distancing policy in advance even before the official announcement of pandemic caused by Covid-19 was made by WHO. Thus, the investors gained confident and hoped that with the reduction number of daily confirmed cases, the business would be allowed to resume operations. However, the interaction effect of social distancing policy and daily growth rate of covid-19 cases revealed that it has negative (positive) effect on the

**Table 10. Robustness tests: Additional control variables.**

| Panel | Daily Covid-19 Death Growth | | | | | | Log (Daily Vaccination) | | | | | |
|---|---|---|---|---|---|---|---|---|---|---|---|---|
| | (1) | (2) | (3) | (4) | (5) | (6) | (1) | (2) | (3) | (4) | (5) | (6) |
| CGRATE | -0.025*** | -0.002 | -0.024*** | 0.020*** | -0.003 | 0.020*** | -0.026*** | 0.002 | -0.025*** | 0.021*** | -0.005 | 0.024*** |
| | (0.001) | (0.917) | (0.002) | (0.003) | (0.849) | (0.001) | (0.001) | (0.906) | (0.002) | (0.002) | (-0.716) | (0.001) |
| SI | 0.014* | 0.018** | | 0.009* | 0.005 | | 0.011 | 0.018*** | | 0.012** | 0.006 | |
| | (0.099) | (0.018) | | (0.059) | (0.245) | | (0/134) | (0.008) | | (0.022) | (0.221) | |
| CGRATE*SI | | -0.001 | | | 0.001 | | | -0.001* | | | 0.001** | |
| | | (0.250) | | | (0.109) | | | (0.051) | | | (0.029) | |
| C1 | | | 0.044 | | | 0.044 | | | 0.030 | | | 0.024 |
| | | | (0.403) | | | (0.189) | | | (0.520) | | | (0.447) |
| C2 | | | 0.060 | | | 0.037 | | | 0.020 | | | -0.014 |
| | | | (0.474) | | | (0.495) | | | (0.798) | | | (0.794) |
| C3 | | | 0.095 | | | 0.230* | | | -0.018 | | | 0.108 |
| | | | (0.596) | | | (0.061) | | | (0.892) | | | (0.222) |
| C4 | | | 0.006 | | | 0.021 | | | -0.030 | | | -0.019 |
| | | | (0.888) | | | (0.422) | | | (0.448) | | | (0.449) |
| C5 | | | 0.058 | | | -0.010 | | | -0.010 | | | -0.037 |
| | | | (0.567) | | | (0.875) | | | (0.898) | | | (0.498) |
| C6 | | | -0.083 | | | -0.008 | | | -0.093 | | | -0.068 |
| | | | (0.327) | | | (0.876) | | | (0.260) | | | (0.215) |
| C7 | | | 0.188 | | | -0.011 | | | 0.204* | | | 0.059 |
| | | | (0.101) | | | (0.868) | | | (0.082) | | | (0.440) |
| C8 | | | 0.109 | | | 0.176*** | | | 0.120 | | | 0.170*** |
| | | | (0.194) | | | (0.004) | | | (0.233) | | | (0.006) |
| Additional var. | -0.004 | -0.002 | -0.005 | 0.004 | 0.001 | 0.004 | 0.009 | 0.004 | -0.019 | -0.023* | -0.019 | -0.028** |
| | (0.272) | (0.629) | (0.297) | (0.212) | (0.645) | (0.222) | (0.649) | (0.839) | (0.233) | (0.064) | (0.124) | (0.014) |
| Control var. | Yes | Yes | Yes | Yes | Yes | Yes | Yes | Yes | Yes | Yes | Yes | Yes |
| Obs. | 575 | 575 | 575 | 575 | 575 | 575 | 575 | 575 | 575 | 575 | 575 | 575 |
| $R^2$ | 0.109 | 0.115 | 0.115 | 0.191 | 0.204 | 0.220 | 0.103 | 0.114 | 0.080 | 0.185 | 0.205 | 0.185 |
| Adjusted $R^2$ | 0.095 | 0.099 | 0.090 | 0.178 | 0.190 | 0.198 | 0.088 | 0.099 | 0.064 | 0.172 | 0.191 | 0.171 |
| F-stat (p-value) | 7.609*** | 7.340*** | 4.562*** | 14.875*** | 14.453*** | 9.872*** | 7.201*** | 7.294*** | 4.912*** | 14.252*** | 14.066*** | 12.853*** |

This table presents the results of different robustness. The regression results use $RET_{FBMKLCI}$ and $VOL_{FBMKLCI}$ as dependent variables in Column 1–3 and Column 4–6, respectively, and after including two variables-one at a time-as additional variables to the main regressions. Daily COVID-19 death growth rate is the daily growth rate of COVID-19 death cases in Malaysia. Log (Daily Vaccination) is the natural logarithm of the daily number of COVID-19 vaccinations. The number in the brackets are p-value and asterisks denote *, ** and *** statistical significance at 10%, 5% and 1% levels, respectively.

stock market return (volatility). The indirect impact from the interaction terms seems to be consistent and legitimate for similar reason—inconsistent government policies at curbing the coronavirus in the context of Malaysia. The study also suggests that increase in stock volatility can be explained by the restriction in international travel policy as well as cancellations of public events which hurt the domestic economy in Malaysia and thus, the reopening international border to outsiders became unpredictable as it was highly dependent on the virus condition in the country. It is also acknowledged that policy implication from social distancing measure has diverse impact on different sectors. Hence, policymakers and government should not hesitate to implement the non-pharmaceutical policies in combating the infection and mortality rate as it shows a good governance of country during the pandemic. Subsequently, this will

**Table 11. Robustne ss tests: Additional control variables.**

| Panel | With weekday dummies | | | | | |
|---|---|---|---|---|---|---|
| | **(1)** | **(2)** | **(3)** | **(4)** | **(5)** | **(6)** |
| CGRATE | -0.023*** | -0.004 | -0.023*** | 0.022*** | -0.006 | 0.022*** |
| | (0.000) | (0.780) | (0.003) | (0.002) | (0.659) | (0.001) |
| SI | 0.011 | 0.016** | | 0.011** | 0.005 | |
| | (0.125) | (0.018) | | (0.030) | (0.273) | |
| CGRATE*SI | | -0.001 | | | 0.001** | |
| | | (0.209) | | | (0.026) | |
| C1 | | | 0.038 | | | 0.049 |
| | | | (0.407) | | | (0.131) |
| C2 | | | 0.038 | | | 0.051 |
| | | | (0.616) | | | (0.348) |
| C3 | | | 0.075 | | | 0.251** |
| | | | (0.665) | | | (0.039) |
| C4 | | | 0.003 | | | 0.021 |
| | | | (0.949) | | | (0.435) |
| C5 | | | 0.046 | | | 0.000 |
| | | | (0.592) | | | (0.998) |
| C6 | | | -0.085 | | | -0.010 |
| | | | (0.295) | | | (0.850) |
| C7 | | | 0.177 | | | -0.002 |
| | | | (0.114) | | | (0.982) |
| C8 | | | 0.112 | | | 0.178** |
| | | | (0.172) | | | (0.005) |
| $D_2$ | 0.484*** | 0.443*** | 0.489*** | -0.003 | 0.056 | 0.003 |
| | (0.000) | (0.000) | (0.000) | (0.967) | (0.515) | (0.972) |
| $D_3$ | 0.407*** | 0.357*** | 0.410*** | -0.077 | -0.005 | -0.073 |
| | (0.001) | (0.004) | (0.000) | (0.368) | (0.954) | (0.375) |
| $D_4$ | 0.347*** | 0.306*** | 0.350*** | -0.010 | 0.050 | -0.012 |
| | (0.003) | (0.013) | (0.004) | (0.904) | (0.550) | (0.877) |
| $D_5$ | 0.305** | 0.262* | 0.307** | -0.007 | 0.055 | -0.006 |
| | (0.021) | (0.065) | (0.017) | (0.943) | (0.612) | (0.949) |
| Control var. | Yes | Yes | Yes | Yes | Yes | Yes |
| Obs. | 575 | 575 | 575 | 575 | 575 | 575 |
| $R^2$ | 0.132 | 0.137 | 0.138 | 0.183 | 0.205 | 0.213 |
| Adjusted $R^2$ | 0.113 | 0.117 | 0.109 | 0.166 | 0.186 | 0.186 |
| F-stat (p-value) | 7.106*** | 6.849*** | 4.693*** | 10.505*** | 11.126*** | 7.933*** |

This table presents the results of different robustness. The regression results use $RET_{FBMKLCI}$ and $VOL_{FBMKLCI}$ as dependent variables in Column 1–3 and Column 4–6, respectively, and after including weekday dummies as additional variables to the main regressions. $D_2$, $D_3$, $D_4$ and $D_5$ are weekday dummies which take on the value 1 if the corresponding return for that particular day is Tuesday, Wednesday, Thursday and Friday, respectively and 0 otherwise. The number in the brackets are p-value and asterisks denote *, ** and *** statistical significance at 10%, 5% and 1% levels, respectively.

gain investors' trust and eventually, stabilizes the financial market. Finally, market participants can utilise the important information such as government policy responses to Covid-19 to restructure their portfolio during the pandemic.

## Author Contributions

**Conceptualization:** Racquel Rowland.

**Formal analysis:** Racquel Rowland.

**Investigation:** Ricky Chee Jiun Chia.

**Methodology:** Racquel Rowland.

**Project administration:** Ricky Chee Jiun Chia.

**Supervision:** Venus Khim-Sen Liew.

**Writing – review & editing:** Racquel Rowland, Ricky Chee Jiun Chia.

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
