## [Decision Letter · Decision Letter 0]

30 May 2022

PONE-D-22-08600Do non-pharmaceutical policies work in response to COVID-19? Evidence from Malaysia Stock Market Return and VolatilityPLOS ONE

Dear Dr. Chia,

Thank you for submitting your manuscript to PLOS ONE. After careful consideration, we feel that it has merit but does not fully meet PLOS ONE’s publication criteria as it currently stands. Therefore, we invite you to submit a revised version of the manuscript that addresses the points raised during the review process.

We look forward to receiving your revised manuscript.

Kind regards,

Aurelio F. Bariviera, Ph.D.

Academic Editor

PLOS ONE

Journal Requirements:

2. PLOS ONE does not copy edit accepted manuscripts (https://journals.plos.org/plosone/s/criteria-for-publication#loc-5). To that effect, please ensure that your submission is free of typos and grammatical errors.

Additional Editor Comments:

I recommend the authors to read carefully the reviewers' comments.

In addition, I would like to suggest to better explain their contribution to the empirical literature. There are several papers analyzing the effect of Covid-19 and mitigation policies on stock markets, bond markets, cryptocurrencies, etc. Therefore, authors could summarize some papers, explaining findings and trying to find some common pattern with previous research.

I think the definition of volatility is not correct. The authors define it as: log|Ret|_t. However, if RET=0.5, then log|RET| will be negative. This could be solved just by redefining RET=log(P_t-Pt-1), and then VOL=|RET|. Alternatively, you could define it as |log|RET||

Finally, and in line with Reviewer 1, I suggest to update data and results.

Reviewers' comments:

Reviewer's Responses to Questions

**Comments to the Author**

1. Is the manuscript technically sound, and do the data support the conclusions?

Reviewer #1: Yes

Reviewer #2: Yes

2. Has the statistical analysis been performed appropriately and rigorously? 

Reviewer #1: Yes

Reviewer #2: Yes

3. Have the authors made all data underlying the findings in their manuscript fully available?

Reviewer #1: Yes

Reviewer #2: Yes

4. Is the manuscript presented in an intelligible fashion and written in standard English?

Reviewer #1: Yes

Reviewer #2: Yes

5. Review Comments to the Author

Reviewer #1: The abstract starts with “Before vaccination policy introduced in the early 2021, Malaysian government has made the decision to impose non-pharmaceutical intervention to contain the deadly virus.” I believe there is no need for it as all the countries did the same. Please focus on the aim, data, methodology and findings in the abstract.

This topic has been widely studied both in single country context and in multi-country settings. So what is new in the paper? The paper lacks a contribution. Even if the authors believe that there is a contribution, it is not written in the introduction. How this paper differs from the previous studies? Moreover why Malaysia? Why it is interesting to see single country findings while we have studies using multi-country data? Simply, the paper lacks a contribution. Same topics have been studied before. It is a replication for Malaysia.

In the introduction, authors say that: “With the inconsistent standard operating system going on in Malaysia, the accumulated number of COVID-19 confirmed cases is accelerated at exponential rate on a daily basis”. There is no exponential growth of cases now. Please pay attention to your arguments.

The starting period of the data is clarified but the ending of the data period is not clear. Why the data ends on 30th November, 2021? I am not asking for a data from May 2022 but it is now 6 months old and if and when it is published, it will be 8 months old (at least).

The use of the interaction of growth in Covid-19 case and SI is not clear to me.

There is a problem with the use of notations, subscripts.

Do you consider the autocorrelation?

The paper lacks robustness checks. You can use different measures of volatility. You can use growth of deaths instead of growth of cases.

Do you control for the day of the week effect?

You can also check if different industries are differently affected.

Vaccination should be considered as it becomes an important factor.

Reviewer #2: Accept in its present form. The study fills the gap in the literature and provide new evidence on the measures undertaken by the government and its impact on the local stock market. Data and methodology is sufficiently described and explained.

6. PLOS authors have the option to publish the peer review history of their article (what does this mean?). If published, this will include your full peer review and any attached files.

Reviewer #1: No

Reviewer #2: No

---

## [Author Response · Author response to Decision Letter 0]

9 Aug 2022

Rebuttal Letter

We thank the editor and the two viewers for their comments on our manuscript. Below is our response to each point raised by the academic editor and reviewers. We hope that we satisfyingly addressed them and that the manuscript will be now suited for publication.

Sincerely,

On behalf of all authors,

Racquel Rowland

Academic editor:

The manuscript has been altered to adhere to PLOS ONE style requirement.

2. PLOS ONE does not copy edit accepted manuscripts.

The manuscript was presented and written in Standard English.

Additional Editor Comments:

I recommend the authors to read carefully the reviewers' comments. In addition, I would like to suggest to better explain their contribution to the empirical literature. There are several papers analyzing the effect of Covid-19 and mitigation policies on stock markets, bond markets, cryptocurrencies, etc. Therefore, authors could summarize some papers, explaining findings and trying to find some common pattern with previous research.

The empirical finding from previous papers has been updated in the empirical literature and can be seen on page 5, lines 99-135. 

I think the definition of volatility is not correct. The authors define it as: log|Ret|_t. However, if RET=0.5, then log|RET| will be negative. This could be solved just by redefining RET=log(P_t-Pt-1), and then VOL=|RET|. Alternatively, you could define it as |log|RET||.

The measurement of volatility was re-adjusted and re-calculated based on the correct definition suggested by the editor and can be seen on page 5, line 186.

Finally, and in line with Reviewer 1, I suggest to update data and results.

The data has now updated until 31st May 2022 at page 1, line 17 and page 5, line 162 whereas new results have generated and referred to page 6-10, lines 221-393.

Reviewer #1

The abstract starts with “Before vaccination policy introduced in the early 2021, Malaysian government has made the decision to impose non-pharmaceutical intervention to contain the deadly virus.” I believe there is no need for it as all the countries did the same. Please focus on the aim, data, methodology and findings in the abstract.

The abstract starts with ‘Before vaccination policy introduced in the early 2021, Malaysian government has made the decision to impose non-pharmaceutical intervention to contain the deadly virus…” was removed and amended with the necessary information as requested by reviewer which can refer to page 1, line 15-16.

This topic has been widely studied both in single country context and in multi-country settings. So what is new in the paper? The paper lacks a contribution. Even if the authors believe that there is a contribution, it is not written in the introduction. How this paper differs from the previous studies? Moreover why Malaysia? Why it is interesting to see single country findings while we have studies using multi-country data? Simply, the paper lacks a contribution. Same topics have been studied before. It is a replication for Malaysia.

The explanation on choosing Malaysia stock market as focus of the study is explained in the contribution of study which can be found on page 4 lines 136-151.

In the introduction, authors say that: “With the inconsistent standard operating system going on in Malaysia, the accumulated number of COVID-19 confirmed cases is accelerated at exponential rate on a daily basis”. There is no exponential growth of cases now. Please pay attention to your arguments.

Yes, there was no exponential growth of cases now. However, the sentence was incomplete as the exponential growth of cases was referring to the spike of covid-19 cases during the third wave of pandemic as the Malaysian government started to ease the restriction policies. It was reported that the third wave started in September 2020 which origins from two big clusters, namely the Benteng Lahad Datu cluster,in Sabah state and Kedah’s Tembok cluster. The sudden spike was due to poor compliance with COVID-19 SOPs. Hence, this statement has amended on page 4, line 143-153.

The starting period of the data is clarified but the ending of the data period is not clear. Why the data ends on 30th November, 2021? I am not asking for a data from May 2022 but it is now 6 months old and if and when it is published, it will be 8 months old (at least).

The ending period of the data now has updated to the latest which is until May 31, 2022 as requested by the reviewers and can refer to page 1, line 17 and page 5, line 162. The reason for choosing this date as ending period because first, some of the macroeconomic variables data availability such as interest rate, unemployment rate, industrial production index and consumer price index are only available until 31st May, 2022 and secondly, it is also for synchronization purpose.

The use of the interaction of growth in Covid-19 case and SI is not clear to me.

The moderation effect generated from the interaction variables will be able to explain how the stringency index which proxied for social distancing measure enable to moderate the effect of covid-19 growth case on the stock return and volatility. As the government impose stringent social distancing measure such as lockdown and restriction movement were introduced to combat the disease, it is believed that such policy will reduce the infection rate of covid-19. Following the policy implementation, the investor is able to restructure their portfolio accordingly. The explanation on the use of interaction variable of COVID-19 growth rate and stringency index is amended and can be found on page 6, lines 204-208. 

There is a problem with the use of notations, subscripts.

The amendment is made to the equation on page 5, lines 184 and page 6, line 201, 213 with the correct use of notation and subscripts.

Do you consider the autocorrelation?

This study bootstraps the standard errors for the OLS coefficients, using the empirical residuals. As such, the resulting t-value and marginal significance value (p) are valid regardless of the nature of the residuals. 

The paper lacks robustness checks. You can use different measures of volatility. You can use growth of deaths instead of growth of cases.

Good point. The daily COVID-19 death growth variable will give an additional information for market participants in repositioning their portfolio since the resumption of economic activity relies heavily on how well the government in handling the deadly disease by preventing the daily death growth from accelerating. Hence, the employment of daily covid-19 death growth is introduced to all three model specifications and its result can refer to page 20, Panel A in Table 10. The discussion of the empirical findings can be found on page 9, lines 355-362 

Do you control for the day of the week effect?

The weekday dummies variable is introduced as additional variable to all three model specifications in order to control for day-of-the-week and the explanation of the effect can be found on page 10 lines 268-371 and the results can be found on page 21, Table 11.

You can also check if different industries are differently affected.

Good point. As the government imposed social distancing measure, many sectors have severely affected by the policies which disrupt the economic activity. Thus, the use of different industries or sectors was introduced as dependent variables to capture the diverse impact in response to the non-pharmaceutical intervention by government. Hence, the results and discussion for the policy implication on different sectors can refer to Table 4-10 on page 14-19 and page 9, lines 320-348.

Vaccination should be considered as it becomes an important factor.

Good point. The employment of vaccination variable into the model specification enables to identify which policies works best at mitigating the disease by comparing it with the non-pharmaceutical strategy proxy in determining the stock return as well as volatility. Hence, the log (daily vaccination) variable is introduced as additional variable to all three model specifications with alternative dependent measure. The results and discussion can be referred to page 9-10, lines 359-368 and page 20, Panel B in Table 10 respectively.

---

## [Decision Letter · Decision Letter 1]

23 Aug 2022

PONE-D-22-08600R1Do non-pharmaceutical policies in response to COVID-19 affect Stock Performance? Evidence from Malaysia Stock Market Return and VolatilityPLOS ONE

Dear Dr. Chia,

Thank you for submitting your manuscript to PLOS ONE. After careful consideration, we feel that it has merit but does not fully meet PLOS ONE’s publication criteria as it currently stands. Therefore, we invite you to submit a revised version of the manuscript that addresses the points raised during the review process.

We look forward to receiving your revised manuscript.

Kind regards,

Aurelio F. Bariviera, Ph.D.

Academic Editor

PLOS ONE

Additional Editor Comments:

In addition to the opinion of Reviewer 1, the paper needs a thorough proofreading. Please make the paper be revised by a professional in English language, before resubmitting.

Reviewers' comments:

Reviewer's Responses to Questions

**Comments to the Author**

1. If the authors have adequately addressed your comments raised in a previous round of review and you feel that this manuscript is now acceptable for publication, you may indicate that here to bypass the “Comments to the Author” section, enter your conflict of interest statement in the “Confidential to Editor” section, and submit your "Accept" recommendation.

Reviewer #1: All comments have been addressed

2. Is the manuscript technically sound, and do the data support the conclusions?

Reviewer #1: Partly

3. Has the statistical analysis been performed appropriately and rigorously? 

Reviewer #1: No

4. Have the authors made all data underlying the findings in their manuscript fully available?

Reviewer #1: No

5. Is the manuscript presented in an intelligible fashion and written in standard English?

Reviewer #1: No

6. Review Comments to the Author

Reviewer #1: The authors did a great effort for revising the paper. Data is also updated.

To use “On the other hand”, you need to use on the one hand, please drop this from the abstract.

Please add the citations for the variables you used in the analysis.

Robustness tests add value to the paper. It is important to see if the findings for Malaysia is different from other countries. In this sense, authors can cite some additional papers for comparison.

Please check the citation and reference match, Rouatbi et al. (2021) is cited but not in the references.

This is a misunderstanding of the robustness idea. For day of the week effect, you introduce those day dummies to show if your main findings remain the same or not, not to comment on day of week effect. And we see that CGRATE*SI becomes insignificant. So, your findings are not robust to alternative measures.

“Similarly, a one percent increase 356 in daily covid-19 death growth will trigger the volatility from 0.001% to 0.004% but not statistically significant.” If a coefficient is not significant, it has no effect so there is no need to mention triggering effect. Simply, there is no significant effect of covıd-10 death growth.

“This can explain that daily covid-19 death growth is less significant in explaining the stock market returns and volatility in the context of Malaysia” I couldn’t understand this, death is not significant, it is not less significant. Please read the paper careful and use proper wordings. The paper needs a proofing, a real one.

7. PLOS authors have the option to publish the peer review history of their article (what does this mean?). If published, this will include your full peer review and any attached files.

Reviewer #1: No

---

## [Author Response · Author response to Decision Letter 1]

19 Oct 2022

Rebuttal Letter

We thank the editor and the reviewer for their comments on our manuscript. Below is our response to each point raised by the academic editor and reviewers. We hope that we satisfyingly addressed them and that the manuscript will be now suited for publication.

Sincerely,

On behalf of all authors,

Racquel Rowland

Reviewer #1

To use “On the other hand”, you need to use on the one hand, please drop this from the abstract.

The use “on the other hand” was removed and changed to “on the one hand’ in the abstract which can refer to line 21 on page 1.

Please add the citations for the variables you used in the analysis.

The citation for the additional variables in the model specification was amended and can refer to line 371 to 397 page 10.

Robustness tests add value to the paper. It is important to see if the findings for Malaysia is different from other countries. In this sense, authors can cite some additional papers for comparison.

The amendment can refer to line 371 to 397 page 10.

Please check the citation and reference match, Rouatbi et al. (2021) is cited but not in the references.

The reference for the citation was amended and can refer to line 45 at page 23.

This is a misunderstanding of the robustness idea. For day of the week effect, you introduce those day dummies to show if your main findings remain the same or not, not to comment on day of week effect. And we see that CGRATE*SI becomes insignificant. So, your findings are not robust to alternative measures.

The robustness checks on the social distancing measure indicators was amended in the analysis when weekday dummies was introduce to the model specification and can refer to line 393 to 397 at page 11. 

“Similarly, a one percent increase 356 in daily covid-19 death growth will trigger the volatility from 0.001% to 0.004% but not statistically significant.” If a coefficient is not significant, it has no effect so there is no need to mention triggering effect. Simply, there is no significant effect of covıd-10 death growth. “This can explain that daily covid-19 death growth is less significant in explaining the stock market returns and volatility in the context of Malaysia” I couldn’t understand this, death is not significant, it is not less significant. Please read the paper careful and use proper wordings. The paper needs a proofing, a real one.

The sentence started with ““Similarly, a one percent increase 356 in daily covid-19 death growth will trigger the volatility from 0.001% to 0.004% but not statistically significant.” was removed from the paragraph as suggested by the reviewer.

---

## [Decision Letter · Decision Letter 2]

24 Oct 2022

Do non-pharmaceutical policies in response to COVID-19 affect Stock Performance? Evidence from Malaysia Stock Market Return and Volatility

PONE-D-22-08600R2

Dear Dr. Chia,

We’re pleased to inform you that your manuscript has been judged scientifically suitable for publication and will be formally accepted for publication once it meets all outstanding technical requirements.

Kind regards,

Aurelio F. Bariviera, Ph.D.

Academic Editor

PLOS ONE

Additional Editor Comments (optional):

Reviewers' comments:

Reviewer's Responses to Questions

**Comments to the Author**

1. If the authors have adequately addressed your comments raised in a previous round of review and you feel that this manuscript is now acceptable for publication, you may indicate that here to bypass the “Comments to the Author” section, enter your conflict of interest statement in the “Confidential to Editor” section, and submit your "Accept" recommendation.

Reviewer #1: All comments have been addressed

2. Is the manuscript technically sound, and do the data support the conclusions?

Reviewer #1: Yes

3. Has the statistical analysis been performed appropriately and rigorously? 

Reviewer #1: Yes

4. Have the authors made all data underlying the findings in their manuscript fully available?

Reviewer #1: Yes

5. Is the manuscript presented in an intelligible fashion and written in standard English?

Reviewer #1: Yes

6. Review Comments to the Author

Reviewer #1: The authors considered my final comments. I believe the paper can be accepted. Thank you for their efforts.

7. PLOS authors have the option to publish the peer review history of their article (what does this mean?). If published, this will include your full peer review and any attached files.

Reviewer #1: No
